# In-Vivo Bactericidal Potential of *Mangifera indica* Mediated Silver Nanoparticles against *Aeromonas hydrophila* in *Cirrhinus mrigala*

**DOI:** 10.3390/biomedicines11082272

**Published:** 2023-08-15

**Authors:** Muhammad Akram Raza, Zakia Kanwal, Saira Riaz, Maira Amjad, Shafqat Rasool, Shahzad Naseem, Nadeem Abbas, Naushad Ahmad, Suliman Yousef Alomar

**Affiliations:** 1Centre of Excellence in Solid State Physics, University of the Punjab, Lahore 54590, Pakistan; saira.cssp@pu.edu.pk (S.R.); shafqatcssp.pu@gmail.com (S.R.); shahzad.cssp@pu.edu.pk (S.N.); 2Department of Zoology, Lahore College for Women University, Jail Road, Lahore 54000, Pakistan; zakia.kanwal@lcwu.edu.pk; 3Department of Physics, Clarkson University, Potsdam, NY 13699, USA; amjad@clarkson.edu; 4Department of Chemistry, University of Leicester, Leicester LE1 7RH, UK; na374@leicester.ac.uk; 5Department of Chemistry, College of Science, King Saud University, Riyadh 11451, Saudi Arabia; anaushad@ksu.edu.sa; 6Zoology Department, College of Science, King Saud University, Riyadh 11451, Saudi Arabia

**Keywords:** green synthesis, silver nanoparticles, *Mangifera indica* (Mango) leaves, major indian carp, antibacterial activity

## Abstract

The present study reports the green synthesis of silver nanoparticles from leaves’ extract of *Mangifera indica* (*M. indica*) and their antibacterial efficacy against *Aeromonas hydrophila* (*A. hydrophila*) in *Cirrhinus mrigala* (*C. mrigala*). The prepared *M. indica* mediated silver nanoparticles (Mi-AgNPs) were found to be polycrystalline in nature, spherical in shapes with average size of 62 ± 13 nm. *C. mrigala* (n = ±15/group) were divided into six groups i.e., G1: control, G2: *A. hydrophila* challenged, G3: *A. hydrophila* challenged + Mi-AgNPs (0.01 mg/L), G4: *A. hydrophila* challenged + Mi-AgNPs (0.05 mg/L), G5: *A. hydrophila* challenged + Mi-AgNPs (0.1 mg/L) and G6: *A. hydrophila* challenged + *M. indica* extract (0.1 mg/L). Serum biochemical, hematological, histological and oxidative biomarkers were evaluated after 15 days of treatment. The liver enzyme activities, serum proteins, hematological parameters and oxidative stress markers were found to be altered in the challenged fish but showed retrieval effects with Mi-AgNPs treatment. The histological analysis of liver, gills and kidney of the challenged fish also showed regaining effects following Mi-AgNPs treatment. A CFU assay from muscle tissue provided quantitative data that Mi-AgNPs can hinder the bacterial proliferation in challenged fish. The findings of this work suggest that *M. indica* based silver nanoparticles can be promising candidates for the control and treatment of microbial infections in aquaculture.

## 1. Introduction

Aquaculture is playing a key role in supplying world food and nutrition needs as about 17% of the animal source protein comes from aquaculture. Moreover, fish meat is rich in essential minerals and omega-3 fatty acids which have great health benefits [1,2].

In order to meet the growing needs of fish, it’s important to address the issues and challenges faced by fisheries sector, especially diseases caused by fish pathogens. The parasitism and disease outbreaks not only reduce the fish production but also cause serious economic losses to aquaculture globally [3,4]. Different types of pathogens such as bacteria, viruses, fungi and parasites can cause diseases in fish. Both types of fish, freshwater and marine, are affected by pathogenic diseases such as abdominal dropsy, Fin Rot, Lernaeasis, Saprolegniasis, Anoxia and fish Predators. These diseases can cause serious biological effects in fish including, low feed intake, scale loss, protruding eye, hemorrhage and ulcers [5]. Some of the most common fish pathogens belong to genera *Aeromonas, Edwardsiella*, *Flavobacterium*, *Citrobacter*, *Mycobacterium*, *Shewanella*, *Streptococcus*, and *Pseudomonas* [6].

*Aeromonas hydrophila* (*A. hydrophila*), one of the *Aeromonas* species, is a motile, rod-shaped, opportunistic Gram-negative, facultative anaerobic, nonspore forming ubiquitous bacteria. The most common fish disease caused by this bacterium is called ‘Motile Aeromonas Septicemia’ which is also known by different names including ‘Hemorrhagic Septicemia, Ulcer Disease, or Red-Sore Disease. The list of symptoms in fish infected by *A. hydrophila* includes lack of appetite, skin ulcerations, movement abnormalities, pale gills, and swollen appearance. The fish organs such as gills, liver, kidneys, spleen, pancreas, and skeletal muscle are commonly affected by *A. hydrophila*. Furthermore, *A. hydrophila* is an afflictive increase in the list of multidrug resistant bacteria and fish infection caused by *A. hydrophila* is a ‘zoonotic disease’ and may be transferred to the human by infected fish consumption [7,8,9,10,11].

Antibiotics and vaccination are the most common prophylactic strategies being used against *A. hydrophila* challenge. Nevertheless, growing multidrug resistance against bacteria and undesirable residual product gathering in aquatic environment can cause huge outbreaks of epidemics leading to serious losses to aquaculture in particular and to the environment in general. Regarding vaccination, different challenges have to be faced, such as, specific vaccines have to be administered against particular pathogen, limited types of vaccines are available, higher prices of commercial vaccines and laborious vaccine administration methods [5,12]. Therefore, new safer and economical alternatives are need of the hour to boost aquaculture efficaciously.

In the green synthesis, nanomaterials are made by green chemistry-based methods using the natural products such as plants and microorganisms as reducing and stabilizing agents. This method is cost-effective, and eco-friendly as it reduces the use of noxious chemicals. Moreover, in this synthesis, surfaces of nanoparticles can be functionalized with extract biomolecules such as polyatomic alcohols, proteins, organic ligands, and polysaccharides leading to the enhanced antimicrobial efficacy against different microbes [13,14]. Nanotechnology based techniques can be used in different horizons of aquaculture and fisheries such as detection of fish fractions, pond management, fish breading, fish feed development and delivery, fish health management, pollution remediation and molecular imaging. Furthermore, Green synthesized NPs can be used as potential novel tool to enhance management of drugs and limit the spread of fish diseases [15,16]. Many studies have reported the effective performance of green synthesized AgNPs against disease causing microorganism (e.g., anti-*candida* and Acaricides) and Photocatalytic Activities [17,18].

The use of silver in different forms as a medicinal remedy to prevent infection transmission, to treat various maladies including burns, ulcerations, and infected wounds can be found in literature since the ancient civilization. The antibacterial properties of silver are also well documented [19,20]. Silver at the nano-size level (silver nanoparticles) was found to be more effective and efficient as antimicrobial agent, because objects at nano-level exhibit extraordinary properties owing to large surface-to-volume ratio and quantum confinement effects [21]. In this era of alarmingly rising resistance of microbes against various available drugs, silver nanoparticles (AgNPs) emerged themselves as innovative and effective alternatives to the traditional antibiotics. The antimicrobial ability of AgNPs has been established to combat almost all types of pathogens causing infections in human, animals, birds and aquaculture species [22,23].

*Mangifera indica* (*M. indica*) commonly known as Mango, has been used for various medical applications since centuries owing to biological properties including antibacterial, antioxidant, antiviral, anti-inflammatory, antidiabetic and antiallergic effects. Leaves of *M. indica* are considered important because they have a variety of important biomolecules such as phenolic acid, phenolic esters, flavonoids, triterpenoids, thiamine, mangiferin and chinonin [24,25].

*Cirrhinus mrigala* (*C. mrigala*) is an important fish of the Carp family and is a preferred food source in South Asian countries. In the present study, we prepared AgNPs using the leaves’ extract of *M. indicia* and investigated their anti-bacterial efficacy against *A. hydrophila* in *C. mrigala*. Our hypothesis was that *M. indicia* mediated AgNPs can help fight *A. hydrophila* challenge in *C. mrigala* and thereby, can be employed as promising, novel anit-bacterial measures in aquaculture.

## 2. Materials and Methods

### 2.1. Extraction of Mangifera indica Leaves and Green Synthesis of AgNPs

Fresh leaves of *M. indica* were collected from Botanical Garden of the University of the Punjab, Lahore, Pakistan. In the first step, leaves were washed thrice with running tap water followed by thorough rinsing with deionized (DI) water. The cleansed leaves were shade dried for 10 days before grinding into fine powder by electric grinder. 6 g of this powder was mixed in 50 mL of DI water and was heated up to 70 °C under constant stirring (200 rpm). After 45 min of heating and stirring, solution was taken off from the hot plate and let it to cool down to room temperature in order to sediment the residue. The solution was then centrifuged for 15 min at 4000 rpm and supernatant was separated as extract. Finally, extract was collected by filtering it through Whatmann filter paper (Figure 1).

For the biosynthesis of Mi-AgNPs, procedure mentioned by Philip et al. [24,25,26,27,28,29] was followed with some modifications to achieve optimized conditions. Briefly, 30 mL (1 mM) of silver nitrate (AgNO_3_) precursor solution was heated up to 70 °C under constant magnetic stirring (200 rpm). Thereafter, 3.0 mL of the *M. indica* leaves’ extract was added into the precursor solution dropwise. The optimized pH value of the solution was maintained to 7.6. The color of the solution changed with time from transparent to light brown to dark brown at the end. The changing of the solution color is an indication of reduction reaction and formation of stable AgNPs. The solution was then cooled down to room temperature. In Figure 2, different stages occurring during biosynthesis of *M. indica* leaves’ extract mediated AgNPs are shown.

### 2.2. Characterization of Biosynthesized Mi-AgNPs

The prepared Mi-AgNPs were characterized using Ultraviolet-visible (UV-Vis) spectroscopy (Shimadzu, UV-1800, Kyoto, Japan), X-ray diffractometer (XRD) (JSX 3201M, Jeol, Tokyo, Japan), Fourier transform infrared spectroscopy (FTIR) (IRTracer-100, Shimadzu, Japan), Scanning electron microscopy (SEM) and energy dispersive X-ray spectroscopy (EDX) (Nova NanoSEM 450, Thermo Fisher Scientific, Waltham, MA, USA). For SEM analysis, few drops of Mi-AgNPs were cast on a small (1 × 1 cm^2^) clean piece of soda lime glass slide and dried to obtain sufficient amount Mi-AgNPs for SEM and EDX measurements. Prior to conduct SEM, a thin coating of gold was applied to avoid any possible charging effects.

### 2.3. Experimental Organism and Conditions

*C. mrigala* juveniles of average weight 18 + 3.2 g were procured from local fish hatchery and were acclimatized to the lab conditions for two weeks. Fish were handled for all the experimentation procedures, according to the animal welfare regulations of the host university. Fish were kept in 60 L glass aquaria and were fed two times daily on a rate of 3% of its body weight. The feed composition included soybean meal, fish meal, maize, wheat bran, wheat, vegetable oil, vitamin premix and mineral premix in a proportion of 35%, 28%, 15%, 10%, 7%, 4% 0.9% and 0.1% respectively. Water aeration was supplied continuously through air pumps. 30% of the water was exchanged daily and all the wastes were siphoned. The physicochemical properties of water were as: Ammonia 0.1–0.26 mg/L; Nitrate 0.02–0.09 mg/L; Chloride 10.1–10.4 mg/L; pH 6.8–7.9; dissolved oxygen 5.8–7.4 mg/L and temperature 26.8–28.2 °C.

### 2.4. In-Vivo Challenge Assays

For challenge studies, hypervirulent *A. hydrophila* strain was freshly grown overnight at 37 °C in sterilized nutrient broth (HiMedia, Kennett Square, PA, USA). The obtained bacterial culture was centrifuged at 3000 rpm for 15 min. A bacterial pellet was formed that was rinsed several times with sterilized phosphate bufered saline (PBS), having a pH 7.1. The final bacterial pellet was suspended in PBS. A dilution of bacteria corresponding to 1.8 × 10^5^ CFU/mL was prepared and 0.5 mL of this dilution was intramuscularly injected to the juveniles for challenge assays. Afterwards, juveniles were randomly distributed into following six groups in triplicates (n = ±15/group):G1: control (unchallenged),G2: *A. hydrophila* challenged,G3: *A. hydrophila* challenged + Mi-AgNPs-administered in aquaria water on a daily basis at a concentration of 0.01 mg/L),G4: *A. hydrophila* challenged + Mi-AgNPs-administered in aquaria water on a daily basis at a concentration of 0.05 mg/L),G5: *A. hydrophila* challenged + Mi-AgNPs-administered in aquaria water on a daily basis at a concentration of (0.1 mg/L),G6: *A. hydrophila* challenged + *M. indica* extract-administered in aquaria water at a concentration of 0.1mg/L),G7: Mi-AgNPs treated- Mi-AgNPs-administered in aquaria water on a daily basis at a concentration of (0.1mg/L).

Mi-AgNPs and Mi-Extract were freshly prepared each day, sonicated and administrated to the aquaria water at the above-mentioned concentrations. The concentrations of the AgNPs were chosen based on previous literature regarding nanoparticles administration to fish. Furthermore, optimization was also performed through pilot experiments.

### 2.5. Biochemical Parameters

At the 15th day of treatment fish were anesthetized by using clove oil (100 µg/L). Blood was collected from the caudal vein with disposable sterile syringe. For serum preparation blood was allowed to stand at room temperature for one hour and afterwards was centrifuged for 15 min at 3500 rpm. The activities of Alanine aminotransferase (ALT) Aspartate aminotransferase (AST) and Alkaline phosphatase (ALP) were assessed by Kinetic enzyme assays. Total protein (TP) was estimated through Photometric Colorimetric-Biuret method by using total protein kit (Crescent Diagnostics, Jeddah, Saudi Arabia). Albumin kit (Crescent Diagnostics, Jeddah, Saudi Arabia) was used to calculate the amount of Albumin (Alb). Globulin (Glo) was calculated by subtracting Alb from TP. Albumin/Globulin Ratio (A/G) was also calculated.

### 2.6. Blood Parameters

Ethylenediaminetetra cetic acid (EDTA) was added to the fresh blood to stop clotting. For Total leukocyte count (TLC) and Total erythrocyte count (TEC) blood was diluted in Turk’s solution and Toisson’s solution respectively. Cells were counted on hemocytometer. Hemoglobin (Hb) was measured by Cyanmethemoglobin method using Drabkin’s fluid, and absorbance was recorded on the spectrophotometer at 540 nm. Hematocrit (Hct %) was perceived by using microhematocrit tubes. Blood smears from fresh blood were prepared immediately after blood collection for Differential Leukocyte count (DLC). Blood smears were air dried and later were dipped in 95% methanol for cell fixation. Giemsa stain was used for staining smears. The stained smears were studied under light microscope. Number of Lymphocytes (Lym), Monocytes (Mon), Basophils (Bas), Eosinophils (Eos) and Neutrophils (Neu) were documented.

### 2.7. Histopathological Evaluation

Kidney, liver and gill tissues were immediately fixed in 10% buffered formalin to maintain cell structure and shape. Following the fixation tissues were dehydrated by passing through different grades of ethanol, for 15 min each, in ascending order. Finally, they were kept in absolute ethanol. Tissues were then put in parafin wax and sections of 6–7 µm were cut by using a Microtome machine (ERM-2301). Later the sections were washed successively with xylene and ethanol to remove any extra material. The sections were shifted onto glass slides, stained with hematoxylin and eosin (H&E) and were imaged through optical microscope (Trinocular E-200, Nikon Japan Eil-12).

### 2.8. Oxidative Stress Analysis

Liver and kidney tissues were snap-frozen on liquid nitrogen, later they were rinsed with chilled PBS. Tissues were then homogenized with a Tenbroek glass homogenizer and were centrifuged at 4 °C for 15 min at 10,000 rpm. The supernatant was used to measure malondialdehyde (MDA), superoxide dismutase (SOD), catalase (CAT), and glutathione peroxidase (GPx) through spectrophotometer.

### 2.9. CFU Assay

Equal amounts (0.5 mg) of muscle tissues were collected from *A.hydrophila* challenged, *A. hydrophila* + Mi-AgNPs (0.1 mg/L) and *A. hydrophila* + Mi-Extract (0.1 mg/L) groups to compare the bacterial load. The tissues were suspended in sterile PBS (0.5 mL). Equal volume (100 µL) of suspensions was poured and spread on LB agar plates and placed in the incubator overnight at 37 °C. The growth of bacteria was checked on the next day.

### 2.10. Statistical Analysis

Data was analyzed by GraphPad prism software (Version 4, San Diego, CA, USA). Values in the tables and graphs represent averages ± standard error (SE) of three independent experiments. ANOVA was applied to find significant differences between various groups.

## 3. Results and Discussion

### 3.1. Optical Analysis by UV-Vis Spectroscopy

In Figure 3a, the UV-vis spectra of DI water, AgNO_3_, Mi-extract and Mi-AgNPs are shown. In case of DI and AgNO_3_ samples no absorption peaks were observed in the entire wavelength range of 300–900 nm confirming the absence of AgNPs. A series of narrow peaks can be noticed at lower wavelength values below 400 nm in the case of Mi-extract. Similar absorption properties of *M. indica* leaves extract has also been reported in literature [26,27,28,29]. The reason for these peaks is the presence of biomolecules in Mi-extract e.g., polyphenolic compounds and these absorption peaks can be attributed to the potential π → π* transitions [26,27]. A single absorption peak occurring at around 448 nm was the characteristic band of AgNPs due to surface plasmon resonance (SPR). Various colors of colloidal nanoparticles samples appear because of surface plasmon resonance phenomenon [28]. A comparison of Mi-extract and Mi-AgNPs spectra designates that no strong band appeared below 400 nm in the case of Mi-AgNPs. This can be an indication that the extract biomolecules served as reducing as well as stabilizing agents during the biosynthesis of Mi-AgNPs [29].

### 3.2. Structural Analysis of Mi-AgNPs by XRD

The XRD pattern of Mi-AgNPs showed four noticeable diffraction peaks at 2θ values of 38.25° (111), 45.85° (200), 64.75° (220), and 77.3° (311) (Figure 3b). These peaks can be attributed to Bragg reflection from (111), (200), (200), and (311) planes, respectively, for FCC crystalline silver nature of synthesized Mi-AgNPs (COD ID No. 9313046) [30]. The presence of few extra peaks denoted by stars (*) at 20.8°, 23.95°, and 33.9° can be owed to the crystallization of Mi-extract biomolecules on the surface of Mi-AgNPs.

### 3.3. FTIR Analysis of Mi-Extract and Mi-AgNPs

FTIR is prominent techniques for the identification of different functional groups and bonds. It can be utilized to study the potential biomolecules of Mi-extract involved in the bioreduction and stabilization of Mi-AgNPs [31].

In FTIR spectra major absorption bands appeared around 3271, 2930, 2359, 1653, 1558, 1387, 1362, 1126, 1020, and 929 cm^−1^ (Figure 3c). Almost similar trend in both bands appeared with some changes in peak position and intensity. The broad band around 3271 cm^−1^ can be attributed to O-H functional group owing to the alcoholic, phenolic, and carboxylic acid compounds of the extract. The absorption band around 2903 cm^−1^ can be designated to C-H alkaline and C=C aromatic groups. The string band around 2359 cm^−1^ can be attributed to C≡N nitriles and –C≡C– alkynes. The prominent bands at 1653–1558 cm^−1^ can be assigned to the bending vibration of N–H of 1° amines and stretching vibration of C–C aromatics groups. The bands around 1387–1362 cm^−1^ might be due the presence of rocking vibrations of C-H amide II groups and residue of NO_3_. The strong and narrow bands occurring in the range of 1126–1020 cm^−1^ can be owed to stretching vibrations of C-N of aliphatic amines. The absorption bands between 950–910 cm^−1^ can be ascribed to bending vibrations of O-H of carboxylic acids whereas broad band at 680–500 cm^−1^ can be designated to the aliphatic chain regions of the extract biomolecules [32,33,34,35].

Overall, the FTIR spectrum confirmed the presence of various functional group pertaining to phytochemicals such as flavonoids, polyphenols amines, alcohols, amides and carboxylic acids [15]. The slight shifts in the absorption band positions and intensities are indication that Mi-AgNPs were synthesized by the reduction and stabilization of Mi-extract biomolecules [25,31,36].

### 3.4. Morphological and Compositional Analysis of Mi-AgNPs

SEM analysis revealed that Mi-AgNPs were generally almost spherical in shapes with a size ranging from 50 nm to 75 nm (Figure 4a). The average nanoparticle size was measured to be 62 ± 13 nm ((Figure 4b). Mi-AgNPs appeared sticky to each other perhaps due to the evaporation effects of colloidal Mi-AgNPs droplets on the glass slides.

The elemental composition of Mi-AgNPs was investigated by EDX (Figure 4c). A prominent silver signal (12.05%) around 3keV in the form of intense SPR absorption band confirmed the metallic nature of Mi-AgNPs [37]. Other signals such as C, O, Na, Mg, Al, Si, Ca, Cl, K and Au were also observed. The strong Si signal are probably due to glass slides onto which Mi-AgNPs were mounted. The weak signal of Au was there due to thin gold castings applied on the sample before SEM to minimize the charging effects. The occurring of intense O and C bands can be attributed to the Mi-extract biomolecules deposited on the surfaces of Mi-AgNPs. [37,38].

### 3.5. The Potential Mechanism for Biosynthesis of Mi-AgNPs

One of the major concerns about the use of nanotechnology is their toxicity to the animal cells. The advent of green nanotechnology helped to minimize this risk by substituting the use of hazardous reducing and capping agents with the natural products i.e.,the plant extracts. The plant biomolecules not only serve as reducing agents for the bio-reduction of the procures molecules but also provide a capping layer of biomolecules enhancing the stability and biocompatibility of the nanomaterials [25,32,37,39]. In case of Mi-leaves extract, various metabolites such as phenolic acid, phenolic esters, flavonoids, triterpenoids, thiamine, mangiferin and chinonin are present [25,40].

Figure 5 illustrates phenomena occurring at each step during the synthesis of Mi-AgNPs. The production of silver ions (Ag^+^) starts as soon as precursor salt, AgNO_3_ dissolves in DI water. These Ag-ions require some reducing agent to convert onto free Ag-atoms (Ag^0^), in this case metabolites of Mi-leaves extract serve as bio-reducing agent to convert Ag^+^ to Ag^0^. The heating and construal stirring helped in enhancing the reaction rate and developing the suitable reaction condition for the generation of nuclei from the free Ag-atoms under the effect of Brownian motion and van der Waals interactions. As the reaction proceeds, nucleation and growth processes go side by side. Next, Mi-extract biomolecules came into play and start capping the Mi-AgNPs by behaving as stabilizing agents.

### 3.6. Biochemical Analysis

Living organisms have a special defense system, the formation of reactive oxygen species (ROS), that help them to cope with the microbial challenges. The oxidative burst results in various enzymatic and biochemical changes in the host to control the pathogenic spread. Results of the biochemical parameters indicated that the liver enzyme ALT, AST and ALP were increased in the *A. hydrophila* challenged fish. Infection caused the enzyme concentrations to rise significantly in the infected fish (Table 1). However, the treatment of infected fish with higher concentrations (0.05 and 0.1 mg/L) of Mi-AgNPs showed normalization of ALT and AST. Furthermore, levels of TP, Alb and Glo were raised significantly in challenged fish. Challenged group with Mi-AgNPs treatment were comparable to the control group. The highest A/G ratio was observed in case of challenged fish but was comparable to control in all other groups. The Mi- extract also showed positive recovery effects but the results were not as robust as with Mi-AgNPs. The group having only Mi-AgNPs exposure, (maximum concentration: 0.1 mg/L) did not show any significant differences than the control group. Therefore, it was confirmed that Mi-AgNPs did not cause any adverse side-effects of their own and allowed a safe use against *A.hydrophila* challenge in the present study.

Mango kernel has been shown to stimulate immunity in *L. rohita* against *A. hydrophila* challenge [41]. Moreover, immunomodulatory effects of AgNPs phyto-synthesized by using *Azadirachta indica* have been reported in *C. mrigala* fingerlings challenged with *A. hydrophila*. Similar to our finding the effects of AgNPs were reported to be more promising than the extract [42]. The efficacy of biogenic AgNPs to control the biochemical and enzymatic changes are not only limited to animals. Addition of plant based AgNPs prepared from aqueous extract of *Moringa oleifera* leaves resulted in low amounts of SOD, CAT and POD activities in rice plants with biotic stress. AgNPs reduced the biotic stress by adjusting the production of proteins, enzymatic and nonenzymatic compounds and controlled the proliferation of *Aspergillus flavus* [43].

### 3.7. Blood Analysis

The *A. hydrophila* infection caused decline in TLC, TEC, Hb and Hct. Mi-AgNPs showed improvements by recovering these parameters (Table 2). Mi-AgNPs at the concentration of 0.05 and 0.1 mg/L were seen to be more effective than the lower concentration (0.01 mg/L). Mi-AgNPs also enhanced DLC markers, Lym, Mon, Bas, Eos and Neu at all the three concentrations particularly at the highest concentration. Furthermore, the group treated with Mi-AgNPs alone did not show significant differences to the control group indicating their non-toxic behavior on the blood profile in particular. Hematological parameters, being very sensitive to stressors, are informative tool to check the health status of any organism [44]. The reduction in hematological parameters can be ascribed to suppressed immune response and physiological disturbances due to bacterial burden. Significant reduction in blood parameters were found in rainbow trout infected with *Pseudomonas putida*. Previously, it was shown that AgNPs synthesized by wet chemical method hindered the changes in hematological parameters in *Pseudomonas aeruginosa* challenged *L. rohita* [5].

### 3.8. Histopathological Analysis

For histopathological analysis, 0.1 mg/L concentration of Mi-AgNPs was employed, being the most effective concentration in the previous experiments. Same concentration of the extract was also used for this assay for a comparable analysis. In the control group normal tissue morphology of liver, gills and kidney was observed (Figure 6a,f,k). *A. hydrophila* challenged fish showed blood congestion, edema, inflammation, necrosis and vacuolation (Figure 6b,g,l). *A. hydrophila*+Mi-AgNPs treated group showed improved tissue morphology which was comparable to the control group (Figure 6c,h,m). The challenged fish treated with Mi-Extract however, could not control the infection progression and showed symptoms of tissue destruction (Figure 6d,i,n). Fish treated with only Mi-AgNPs show almost normal tissue histology comparable to the control group (Figure 6e,j,o). Microbial infections and trauma often lead to tissue injuries and inflammations [45,46]. The conventional anti-inflammatory drugs (steroids and nonsteroids) have been used but they cause numerous side effects [47,48]. Plant based synthesis and use of nanoparticles has proved to be an effective anti-inflammatory approach [47]. AgNPs synthesized by *Selaginella myosurus* performed anti-inflammatory action under in vivo and in vitro conditions. The study described that AgNPs can inhibit inflammation by antagonizing the action of acute inflammatory mediator proteins (histamines, serotonin, kinins, prostaglandins, and cyclooxygenase) denaturation in the Carrageenan-induced rat hind paw edema model [49]. In another study, AgNPs generated from *Lindera strychnifolia* exhibited wound healing property [50]. Some possible pathways by which green synthesized metallic NPs can reduce inflammation are blocking of pro-inflammatory cytokines, ROS scavenging activity, and inhibition of NF-κB and COX-2 pathways [47].

### 3.9. Oxidative Stress Biomarkers

MDA levels were increased significantly both in the gills and liver of the challenged fish (Table 3). However, Mi-AgNPs suppressed the induction of MDA. The Mi-Extract could not prevent the induction of MDA. Levels of SOD, CAT and GPx were decreased in *A. hydrophila* challenged group. It was noted that Mi-AgNPs restored the levels of these markers in a dose dependent manner. In the group treated with only Mi-AgNPs, no significant differences were found than the control group showing that the prepared Mi-AgNPs did not cause any oxidative burden of their own on the fish. It has been reported that AgNPs prepared from *M. indica* seed aqueous extract perform dose dependent antioxidant activity [51]. Moreover, AgNPs prepared from *Catharanthus roseus* flower extracts also showed stronger antioxidant potential in comparison to its flower extract [52]. Plants are enriched with different phytoconstituents which can perform antioxidant activity due to their redox potential [53]. The higher antioxidant potential of nanoparticles than the extract itself is attributed to better adsorption of the antioxidant materials from the extract onto the surface of the nanoparticles [54]. Thus, the biogenic AgNPs can be a good source of natural antioxidants for controlling oxidative stress caused by microbial infections.

### 3.10. CFU Analysis

Bacterial colonies were counted from the agar plates which showed that Mi-AgNPs (0.1 mg/L) notably decreased the proliferation of bacteria (Figure 7). Bacterial lawns were formed on the plates coated with tissue suspensions from challenged only and challenged +Mi-Extract groups (Figure 7a and Figure 7c respectively). However, countable, lesser number of colonies were formed on the plate coated with challenged + Mi-AgNPs sample (Figure 7b). The CFU analysis can be improved by making serial dilutions of the homogenate samples for countable number of colonies. In a previous study, AgNPs prepared from *M. indica* extract were seen to perform strong antibacterial performance against Gram-negative bacteria viz, *P. aeruginosa K. pneumoniae E. coli* and *S. typhimurium* [51]. In another study AgNPs were synthesized using *Moringa oleifera* leaf extract and showed antimicrobial activity against clinical isolates of *S. aureus*, *C. tropicalis*, *C. krusei* and *K. pneumonia* [55]. AgNPs prepared from the Peel Extract of Mango have been reported to show antibacterial performance against *S. aureus* and *E. coli* [56]. AgNPs prepared using other extracts such as *Lavandula angustofolia, L*. *Otostegia persica*, *Catharanthus roseus*, *Lantana camara* and *Albizia procera* have also been exhibited significant antibacterial effects against *S. aureus, E. coli* and *Pseudomonas aeruginosa* [52,57,58,59]. The potential mechanisms for the demolition of bacteria by nanoparticles are the disruption of the polymeric subunits of cell membrane and cell wall, exudation of the cellular materials and disturbance of the bacterial protein synthesis [60,61].

## 4. Conclusions

Nanotechnology offers numerous benefits for biological applications but the synthesis methods bring forth some concerns. The synthesis of nanoparticles through the utilization of green resources as a substitute for the synthesizing chemicals has started to revolutionize this approach. This study presents an eco-friendly and biocompatible method for the synthesis of AgNPs from *M. indica* plant extract which served as reducing and stabilizing agent. The biosynthesized Mi-AgNPs exhibited significant biochemical, histological and antioxidant potential against *A. hydrophila* in *C. mrigala*. The bacterial proliferation was low in the challenged fish treated with Mi-AgNPs. Overall, AgNPs showed stronger antibacterial potential than the *M. indica* extract. We observed that the highest concentration of the Mi-AgNPs (0.1 mg/L) was most effective in the disease recovery and therefore, it is recommended that for future studies the reported highest concentration or even higher doses should be tested. Based on these findings it is concluded that *M. indica* mediated AgNPs can be promising candidates for antibacterial applications against pathogens in aquaculture. Our study is novel in presenting a first in vivo challenge assay for the anti-bacterial effect of *M. indica* leaves’ extract mediated AgNPs in *C. mrigala* and provides innovative insights of biogenic remedies for disease treatment in aquaculture. This will help in minimizing the use of pernicious drugs and antibiotics in fisheries which by means of bio-magnification in animal tissues can also be a health hazard to humans.

## Figures and Tables

**Figure 1 biomedicines-11-02272-f001:**
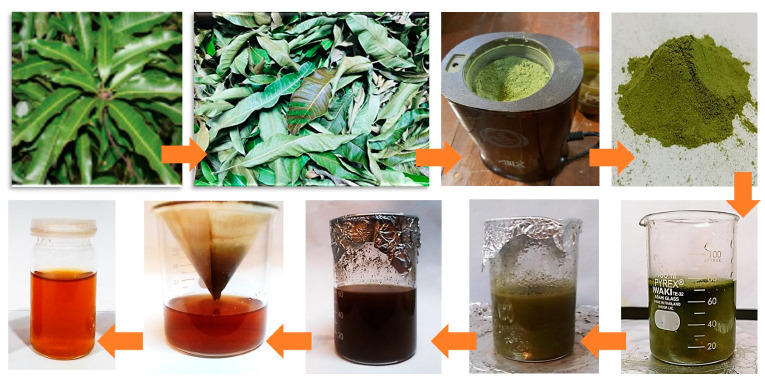
Different steps during the preparation of the *M. indica* leaves’ extract.

**Figure 2 biomedicines-11-02272-f002:**
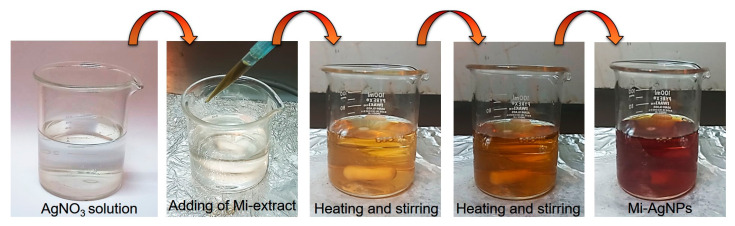
Different stages during the biosynthesis of AgNPs by using the *M. indica* leaves’ extract.

**Figure 3 biomedicines-11-02272-f003:**
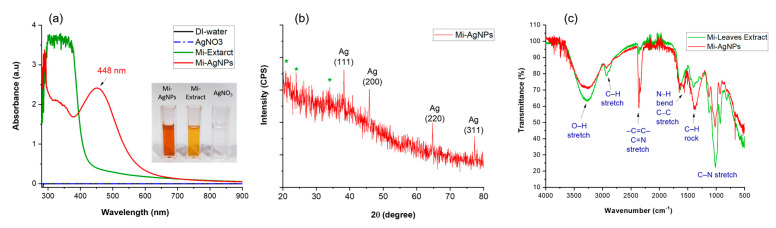
(**a**) UV-Vis spectra of (deionized) DI water, silver nitrate (AgNO_3_), *M. indica* leaves’ extract (Mi-extract) and *M. indica* leaves extract mediated silver nanoparticles (Mi-AgNPs). The inset image exhibits color of different samples in UV cuvettes. (**b**) XRD pattern of Mi-AgNPs, FCC crystalline structure was confirmed by the characteristic Ag-peaks while presence of extract molecules by star peaks. (**c**) FTIR spectra of Mi-extract and Mi-AgNPs indicate the presence of different extract biomolecules on the NPs surfaces.

**Figure 4 biomedicines-11-02272-f004:**
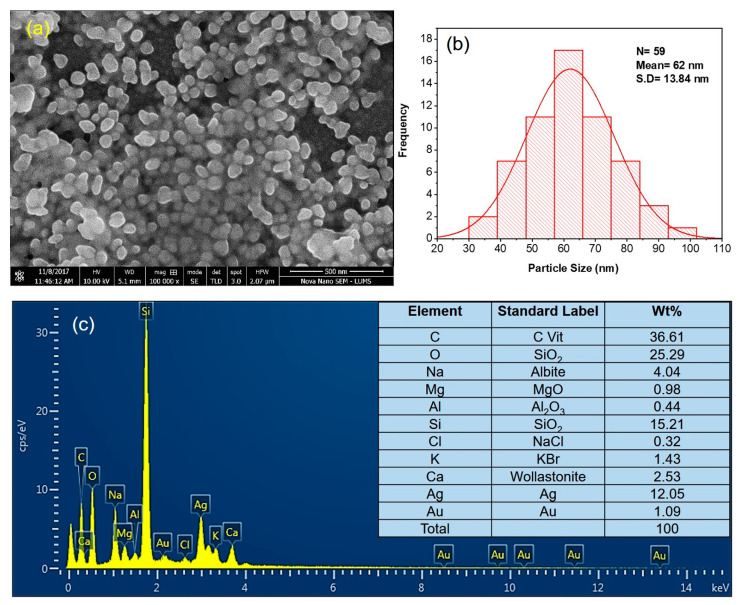
(**a**) SEM micrograph reveals the relatively spherical shape of *M. indica* leaves extract mediated silver nanoparticles (Mi-AgNPs), (**b**) size distribution histogram of Mi-AgNPs indicating the average size AgNPs (62 ±13 nm), (**c**) EDX spectrum of Mi-AgNPs, confirming the presence of Ag as base element and some other Mi-extract molecules which served as reducing agent and stabilizing agent. The inset shows the table of elements indicating amount of each element in weight %.

**Figure 5 biomedicines-11-02272-f005:**
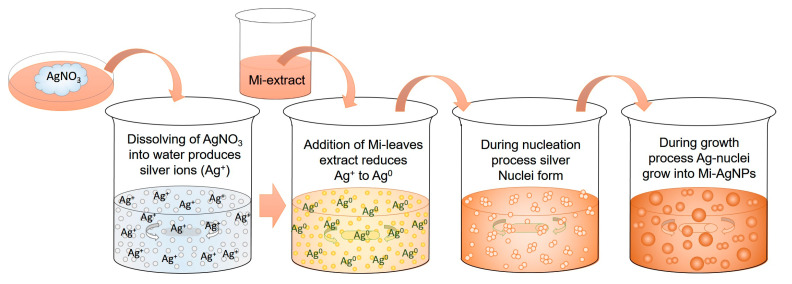
Schematics of the potential mechanism occurring at each step of the biosynthesis of Mi-extract mediated AgNPs.

**Figure 6 biomedicines-11-02272-f006:**
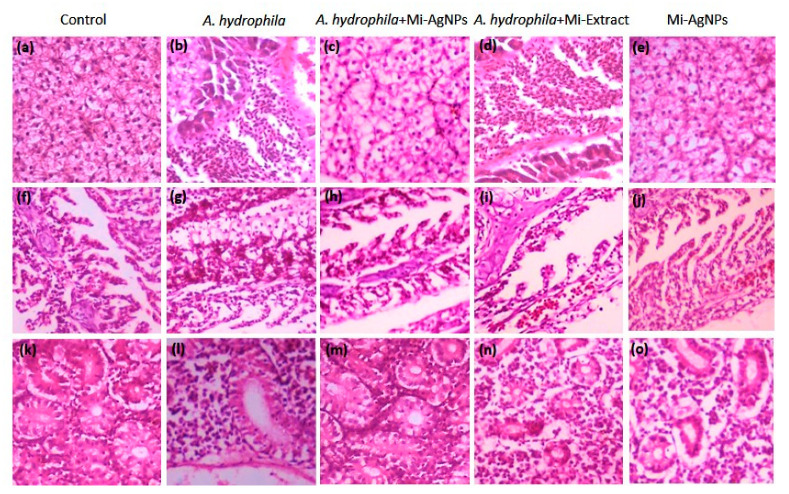
Histopathological evaluation. Liver tissue of Control (**a**), *A.hydrophila* challenged (**b**), *A. hydrophila* challenged +Mi-AgNPs (0.1 mg/L) (**c**), *A. hydrophila*+challenged Mi-Extract (0.1 mg/L) (**d**) and Mi-AgNPs (0.1 mg/L) (**e**). Gill tissue of Control (**f**), *A.hydrophila* challenged (**g**), *A. hydrophila* challenged +Mi-AgNPs (0.1 mg/L) treated (**h**), *A. hydrophila* challenged +Mi-Extract (0.01 mg/L) (**i**) and Mi-AgNPs (0.1 mg/L) (**j**). Kidney tissue of Control (**k**), *A.hydrophila* challenged (**l**) *A. hydrophila* challenged +Mi-AgNPs (0.1 mg/L) (**m**) and *A. hydrophila* challenged +Mi-Extract (0.1 mg/L) (**n**) and Mi-AgNPs (0.1 mg/L) (**o**). (Magnification: 40×).

**Figure 7 biomedicines-11-02272-f007:**
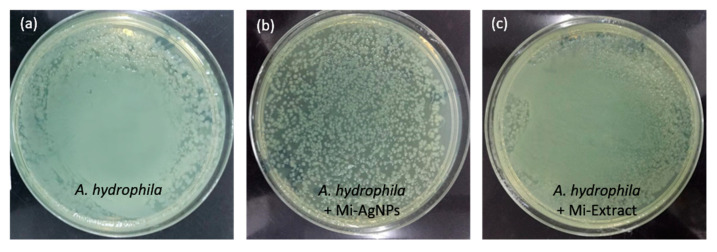
Colony forming unit (CFU) analysis. Homogenate muscle tissues of *A. hydrophila* challenged (**a**), *A. hydrophila* challenged +Mi-AgNPs (0.1 mg/L) (**b**) and *A. hydrophila* challenged +Mi-Extract (0.1 mg/L) (**c**).

**Table 1 biomedicines-11-02272-t001:** Biochemical Parameters. Alanine aminotransferase (ALT) Aspartate aminotransferase (AST) and Alkaline phosphatase (ALP), Total protein (TP), Albumin (Alb), Globulin (Glo), Albumin/Globulin Ratio (A/G). In each parameter the superscripts show significant differences between different groups.

	G1	G2	G3	G4	G5	G6	G7
ALT (U/L	44.5 ± 5.7 ^a^	102.5 ± 11.5 ^b^	105.6 ± 15.6 ^b^	78.1 ± 9.6 ^c^	66.9 ± 5.7 ^a^	95.7 ± 8.5 ^b^	50.1 ± 3.9 ^a^
AST (U/L)	17.3 ± 3.4 ^a^	25.6 ± 2.6 ^b^	24.3 ± 4.9 ^b^	22.7 ± 4.3 ^a^	23.1 ± 4.7 ^b^	27.9 ± 6.4 ^b^	15.6 ± 2.1 ^a^
ALP (U/L)	6.9 ± 1.1 ^a^	78.4 ± 4.6 ^b^	69.2 ± 7.8 ^b^	65.4 ± 10.5 ^b^	45.8 ± 5.1 ^b^	63.5 ± 8.2 ^b^	10.3 ± 4.4 ^a^
TP (g/dL)	3.9 ± 0.6 ^a^	7.8 ± 1.2 ^b^	6.5 ± 1.3 ^b^	7.1 ± 1.6 ^b^	5.8 ± 1.3 ^a^	6.2 ± 1.2 ^b^	3.2 ± 1.1 ^a^
Alb (g/dL)	1.9 ± 0.3 ^a^	4.5 ± 0.8 ^b^	3.1 ± 0.6 ^a^	3.5 ± 0.4 ^a^	3.1 ± 0.6 ^a^	2.9 ± 0.5 ^a^	1.4 ± 0.5 ^a^
Glo (g/dL)	2.0 ± 0.7 ^a^	3.3 ± 0.5 ^b^	3.4 ± 1.1 ^b^	3.6 ± 0.5 ^b^	2.7 ± 0.5 ^a^	3.3 ± 0.7 ^b^	1.8 ± 0.6 ^a^
A/G	0.95 ± 0.1 ^a^	1.3 ± 0.8 ^b^	0.91 ± 0.2 ^a^	0.97 ± 0.08 ^a^	1.14 ± 0.9 ^a^	0.87 ± 0.05 ^a^	1.02 ± 0.3 ^a^

**Table 2 biomedicines-11-02272-t002:** Hematological Parameters. Total leukocyte count (TLC), Total erythrocyte count (TEC), Hemoglobin (Hb), Hematocrit (Hct %), Lymphocytes (Lym), Monocytes (Mon), Basophils (Bas), Eosinophils (Eos), Neutrophils (Neu). In each parameter the superscripts show significant differences between different groups.

	G1	G2	G3	G4	G5	G6	G7
TLC (×10^4^/mm^3^)	220 ± 12 ^a^	123 ± 21^b^	150 ± 19 ^b^	199 ± 11 ^a^	185 ± 24 ^a^	143 ± 16 ^b^	193 ± 2 ^a^
TEC (×10^6^/mm^3^)	156 ± 11 ^a^	78 ± 7 ^b^	98 ± 5 ^b^	123 ± 13 ^a^	132 ± 14 ^a^	9 3 ± 6 ^b^	137 ± 9 ^a^
Hb (g/dl)	9.3 ± 1.4 ^a^	6.7 ± 1.5 ^b^	7.1 ± 1.5 ^b^	7.8 ± 2.4 ^b^	8.7 ± 3.7 ^a^	7.1 ± 1.4 ^b^	9.1 ± 1.7 ^a^
Hct (%)	35 ±3 ^a^	12 ± 1 ^b^	22 ± 2 ^c^	32 ± 3 ^a^	39 ± 5 ^a^	25 ± 6 ^c^	42 ± 5 ^a^
Lym	198 ± 21 ^a^	73 ± 3 ^b^	99 ± 12 ^b^	123 ± 8 ^b^	187 ± 27 ^a^	112 ± 16 ^b^	188 ± 16 ^a^
Mon	86 ± 8 ^a^	34 ± 3 ^b^	71 ± 9 ^a^	85 ± 3 ^a^	91 ± 7 ^a^	56 ± 13 ^b^	75 ± 7 ^a^
Bas	45 ± 3 ^a^	12 ± 2 ^b^	25 ± 4 ^c^	34 ± 3 ^c^	47 ± 4 ^a^	36 ± 8 ^c^	50 ± 5 ^a^
Eos	14 ± 2 ^a^	10 ± 1 ^b^	12 ± 1 ^b^	16 ± 2 ^a^	15 ± 3 ^a^	13.5 ± 2 ^a^	17 ± 4 ^a^
Neu	77 ± 6 ^a^	48 ± 3 ^b^	60 ± 5 ^b^	72 ± 13 ^a^	71 ± 7 ^a^	58 ± 5 ^b^	65 ± 7 ^a^

**Table 3 biomedicines-11-02272-t003:** Oxidative Stress Biomarkers. Malondialdehyde (MDA), Superoxide dismutase (SOD), Catalase (CAT), and Glutathione peroxidase (GPx). In each parameter the superscripts show significant differences between different groups.

	G1	G2	G3	G4	G5	G6	G7
MDA (mol/mg) (Gills)	14.2 ± 2.5 ^a^	43.1 ± 4.2 ^b^	39.9 ± 7.3 ^b^	22.33 ± 0.5 ^a^	19.5 ± 2.6 ^a^	40.7 ± 3.7 ^b^	18.5 ± 1.7 ^a^
MDA (mol/mg) (Liver)	8.3 ± 0.7 ^a^	34.5 ± 1.4 ^b^	23.6 ± 1.1 ^c^	18.5 ± 0.5 ^c^	12.5 ± 0.8 ^a^	30.3 ± 1.4 ^b^	7.8 ± 0.6 ^a^
SOD (U/mg) (Gills)	178.2 ± 11.7 ^a^	90.9 ± 7.9 ^b^	87.6 ± 7.6 ^b^	114 ± 12.5 ^b^	162 ± 7.5 ^a^	72.6 ± 4.7 ^b^	165.4 ± 13.1 ^a^
SOD (U/mg) (Liver)	222.4 ± 15.2 ^a^	162.1 ± 1 7.3 ^b^	187.8 ± 10.6 ^b^	200.9 ± 14.8 ^a^	212 ± 12.5 ^a^	186.3 ± 6.7 ^b^	196.3 ± 23.5 ^a^
CAT (U/mg) (Gills)	56.4 ± 3.6 ^a^	21.5 ± 2.9 ^b^	35.7 ± 5.7 ^b^	45.6 ± 4.7 ^a^	49.7 ± 6.3 ^a^	31.5 ± 2.6 ^b^	62.1 ± 4.9 ^a^
CAT (U/mg) (Liver)	68.7 ± 6.4 ^a^	35.4 ± 4.8 ^b^	55.2 ± 8.8 ^a^	63.9 ± 7.4 ^a^	57.4 ± 3.6 ^a^	48.1 ± 5.8 ^b^	61.7 ± 5.7 ^a^
GPx (U/mg) (Gills)	35 ± 3.5 ^a^	12 ± 1.6 ^b^	22 ± 2.4 ^c^	32 ± 3.8 ^a^	45 ± 5.3 ^a^	25 ± 6.7 ^c^	32.5 ± 4.1^a^
GPx (U/mg) (Liver)	47.7 ± 5.1 ^a^	23.5 ± 2.8 ^b^	28.7 ± 1.2 ^b^	37.2 ± 3 ^c^	41.6 ± 3.6 ^a^	33.5 ± 2.5 ^b^	53.4 ± 6.2 ^a^

## Data Availability

The data will be available upon request.

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
