# Peer review of "In-Vivo Bactericidal Potential of Mangifera indica Mediated Silver Nanoparticles against Aeromonas hydrophila in Cirrhinus mrigala"

_biomedicines, 2023, doi:10.3390/biomedicines11082272_

Round 1

Reviewer 1 Report

The authors investigated Mangifera indica mediated silver nanoparticles against Aeromonas hydrophila in Cirrhinus mrigala and demonstrated that it acts as promising nanoparticle to combat aquaculture pathogens. But I suggest the authors to consider some points discussed below and revise the manuscript accordingly.

Line 22: check the spelling for Mangifera and in the key words too

Line 49-50: no need to italicize the disease names

Line 104: I suggest not to use the anti-proliferative because it usually represents the malignant cells. So use other word instead of anti-proliferative

The authors have to check the spelling for Mangifera throughout the manuscript.

Figure 1: The panels were indicated by alphabets but if you use alphabets it is better to give small explanation in the figure legends. But you already used arrows to indicate the process so there is no need for the alphabets to indicate the figure panels.

Line 122: Please mention the extract name instead of “above prepared extract”

Line 123-124: The authors mentioned that continuous stirring for 30 minutes. I prefer to delete this sentence because there is no specific time, when the solution changes to dark brown. Moreover, authors already mentioned the conditions in the line 121.

Check the figure 2 legend, inappropriate closed bracket after AgNPs and full stop in the middle. The similar comment I gave for figure 1 regarding the alphabets for figure panels and arrows.

In methodology, authors have to mention the strain details of A.hydrophila they used and the challenging methods. Whether the fish was challenged with immersion of intraperitoneal. The section 2.4 have to be more precise. Authors added the AgNPs and extract separately everyday until 15th day. However they mentioned that A.hydrophila+Mi-AgNPs. This is confusing. Authors have to write this section clearly and provide details.

Furthermore, why the authors chose this specific concentrations of AgNPs for this study. Authors have to explain it in the manuscript.

Line 163: check the preposition in this line.

Line 311: what “con” represents here?

Table 1: The level of ALP is 10 times more than the control group. Authors please check the values again in this row.

Figure 6: Authors have to give the group names above the figure panels. Instead of giving repeatedly in the figures please give on the top of the columns. In addition, authors have to check the concentration indicated in the figure legend.

For CFU analysis, authors have to mention which Mi+AgNPs group they had taken for this analysis.

Figure 7: There were so many literatures indicating the antibacterial activity of Mangifera indica. But seeing the figures, there were no differences between control and M.indica extract treated group. Giving detailed discussion related to this point would be advantageous.

Grammar check before publication is advisable. 

Author Response

Response to 1st Reviewer’s Comments

 (Manuscript ID: biomedicines-2509753)

  • The authors investigated Mangifera indica mediated silver nanoparticles against Aeromonas hydrophila in Cirrhinus mrigala and demonstrated that it acts as promising nanoparticle to combat aquaculture pathogens. But I suggest the authors to consider some points discussed below and revise the manuscript accordingly.

  • We are thankful to the learnt reviewer for expert assessment of our manuscript. We are grateful for his/her thoughtful and valuable comments and recommendations to improve our manuscript. All points and issues raised by the learnt reviewer have been considered and the manuscript has been modified accordingly. A detailed point-by-point response to all comments is provided below indicating the implemented changes in the revised version of the manuscript. A highlighted version by ‘Track Changes’ is also included with the resubmission
  • Line 22: check the spelling for Mangifera and in the key words too
  • We are grateful to the reviewer for highlighting this point. The spelling for Mangifera have been checked and corrected throughout the manuscript.

  • Line 49-50: no need to italicize the disease names
  • Reviewer is right, italicized has been removed throughout the manuscript.

  • Line 104: I suggest not to use the anti-proliferative because it usually represents the malignant cells. So use other word instead of anti-proliferative
  • We agree with reviewer recommendation, ‘anti-proliferative’ has been replaced with word ‘anti-bacterial’.

  • The authors have to check the spelling for Mangifera throughout the manuscript.
  • According to reviewer recommendation, we have checked and corrected the spellings throughout the manuscript.

  • Figure 1: The panels were indicated by alphabets but if you use alphabets it is better to give small explanation in the figure legends. But you already used arrows to indicate the process so there is no need for the alphabets to indicate the figure panels.
  • We agree and appreciate the suggestion of the learnt reviewer that the alphabets are unnecessary to show here in this figure. Therefore, the alphabets have been removed from figure 1, 2 and 5. In all other figures, where alphabets are present, a brief explanation of each letter has been provided in the figure legends.

  • Line 122: Please mention the extract name instead of “above prepared extract”
  • According to reviewer’s suggestion “above prepared extract” has been replaced with indica leaves’ extract in the revised version.

  • Line 123-124: The authors mentioned that continuous stirring for 30 minutes. I prefer to delete this sentence because there is no specific time, when the solution changes to dark brown. Moreover, authors already mentioned the conditions in the line 121.
  • We are grateful to the reviewer for the valuable remarks and comments. We agree with the reviewer and the mentioned sentence has been deleted as per recommendation.

  • Check the figure 2 legend, inappropriate closed bracket after AgNPs and full stop in the middle. The similar comment I gave for figure 1 regarding the alphabets for figure panels and arrows.
  • We express our gratitude to the reviewer for the ample assessment of the manuscript. Correction has been done in legend of figure 2, Alphabets have been removed according to the suggestion. Also removed from figure 5 according to the same format.

  • In methodology, authors have to mention the strain details of A.hydrophila they used and the challenging methods. Whether the fish was challenged with immersion of intraperitoneal. The section 2.4 have to be more precise. Authors added the AgNPs and extract separately everyday until 15th day. However they mentioned that A.hydrophila+Mi-AgNPs. This is confusing. Authors have to write this section clearly and provide details.
  • According to reviewer recommendations, the details of the bacteria used, challenging method, Mi-AgNPs and Mi- extract administration has been added in the main text to improve the clarity.

  • Furthermore, why the authors chose this specific concentration of AgNPs for this study. Authors have to explain it in the manuscript.
  • The concentrations of the AgNPs were chosen based on previous literature regarding nanoparticles administration to fish. Furthermore, optimization was also performed through pilot experiments.

  • Line 163: check the preposition in this line.
  • Reply: The pointed preposition has been corrected

  • Line 311: what “con” represents here?
  • The “con” represents control group, we have changed this “con” to complete word “control” in the revised manuscript.

  • Table 1: The level of ALP is 10 times more than the control group. Authors please check the values again in this row.
  • We acknowledge the reviewer for such careful assessment of the manuscript. We have checked the values of ALP and they are written correctly in Table 1.

  • Figure 6: Authors have to give the group names above the figure panels. Instead of giving repeatedly in the figures please give on the top of the columns. In addition, authors have to check the concentration indicated in the figure legend.
  • The names of groups have been given above the figure panels as recommended by the reviewer. There was a typo at one place in the concentration of the extract in the legend, that has also been corrected.

  • For CFU analysis, authors have to mention which Mi+AgNPs group they had taken for this analysis.
  • The information of the Mi-AgNPs taken for CFU analysis has been added in the CFU methodology and results (sections 2.9, and 3.10) and the figure 7 legends. The reason for using this concentration in CFU and histology section was that this proved to be the most effective concentration in other assays of this study.

  • Figure 7: There were so many literatures indicating the antibacterial activity of Mangifera indica. But seeing the figures, there were no differences between control and M.indica extract treated group. Giving detailed discussion related to this point would be advantageous.
  • We fully agree with the learnt reviewer that there are many studies in which antibacterial activity of indica against various gram positive and gram negative strains has been reported [R1-R4]. However, we could find only one report [R5] where antibacterial activity of M. indica leaves extract against Aeromonas hydrophila was discussed. They used higher concentrations [ 1g, 5g and 10g 1 g, 5 g, 10 g mango kernel kg−1 dry diet to Labeo rohita fingerlings). Actually, the antibacterial efficacy of medicinal plant extracts depends on various factor such as part of the plant (leaves, roots, fruit etc), type of solvent (aqueous or alcoholic) in which extract is prepared, concentration of the extract, type of the strain against which activity is assessed. In our case, CFU analysis showed that there were no differences in the control and M. indica leaves extract. The possible reason for this is that we used a very low concentration (0.1 mg/L) of the extract to correspond to the highest concentration of Mi-AgNPs (0.1 mg/L). Therefore, the concentration differences could be the major cause for the nominal antibacterial effect of the extract in our case.

  • Grammar check before publication is advisable. 
  • Reply: We express our gratitude to the reviewer for the ample assessment of the manuscript. We have thoroughly read the revised manuscript and made improvements in scientific and grammatical standards.

At the end, we want to express our sincere gratitude for the insightful comments and positive feedback provided by the reviewer. All the comments and suggestions have tremendously helped to improve our manuscripts quality and aptness.

References:

  • Alaiya, M.A., Odeniyi, M.A. Utilisation of Mangifera indica plant extracts and parts in antimicrobial formulations and as a pharmaceutical excipient: a review. J. Pharm. Sci. 2023, 9, 29. https://doi.org/10.1186/s43094-023-00479-z
  • Maharaj, A.; Naidoo, Y.; Dewir, Y.H.; Rihan, H. Phytochemical Screening, and Antibacterial and Antioxidant Activities of Mangifera indica Leaves. Horticulturae 2022, 8, 909. https://doi.org/10.3390/horticulturae8100909
  • Cardenas, V.; Mendoza, R.; Chiong, L.; Del, Aguila, E.; Alvítez-Temoche, D.; Mayta-Tovalino, F. Comparison of the Antibacterial Activity of the Ethanol Extract vs Hydroalcoholic Extract of the Leaves of Mangifera indica (Mango) in Different Concentrations: An In Vitro Study. J. Contemp. Dent. Pract. 2020, 21(2), 202-206.
  • Hannan, A.; Asghar, S.; Naeem, T.; Ikram Ullah, M.; Ahmed, I.; Aneela, S.; Hussain, S. Antibacterial effect of mango (Mangifera indica) leaf extract against antibiotic sensitive and multi-drug resistant Salmonella typhi. Pak. J. Pharm. Sci. 2013, 26(4), 715-9. PMID: 23811447.
  • Sahu, S.; Das, B.K.; Pradhan, J.; Mohapatra, B.C.; Mishra, B.K.; Sarangi, N. Effect of Magnifera indica kernel as a feed additive on immunity and resistance to Aeromonas hydrophila in Labeo rohita fingerlings, Fish Shellfish Immunol. 2007, 23(1),109-118, https://doi.org/10.1016/j.fsi.2006.09.009 .

Reviewer 2 Report

The paper entitled: “In-vivo Bactericidal potential of Mangifera indica mediated silver nanoparticles against Aeromonas hydrophila in Cirrhinus mrigala” describes the efficacy of plant extract to fabricate AgNPs to biocontrol of bacterial infection caused by A. hydrophila into C. mrigala. The synthesized AgNPs were characterized by UV-vis spectroscopy, FTIR, XRD, SEM, and EDX. Moreover, the experimental design was achieved to investigate the efficacy of different concentrations of AgNPs on fish inoculated with pathogenic bacteria. The manuscript contains promising data and is well organized, but needs major revision before being accepted for publication in Biomedicines.

1-    The hypothesis of the current study should be rephrased to be clearer.

2-    The novelty of the current study should be addressed.

3-    The authors should refer to the efficacy of different biological entities in the green synthesis of Ag-NPs and their promising activities. The following references can be helpful: https://doi.org/10.3390/biom11030341; https://doi.org/10.3390/catal12050462.

4-    In Figure 1, please refer to the meaning of each letter in the Figure legend. This will be applied to each figure containing more panels.

5-    What about the optimization of pH value during NPs synthesis? Please clarify in subsection “2.1”

6-     In subsection “2.2. Characterization of biosynthesized Mi-AgNPs”, the characterization method should be discussed in detail. The following references can be helpful: https://doi.org/10.1016/j.heliyon.2020.e03943; https://doi.org/10.3390/ijms22105096  

7-    In subsection “2.3. Experimental Organism and Conditions”, please clarify how to inoculate A. hydrophila into C. mrigala? Is the author standardized the inoculum?

8-    Please clarify why the authors achieved biochemicals and other analyses after 15 days of treatment. What happens if achieved after interval times?

9-    How to detect the bacteria calculated in subsection 3.10 is A. hydrophila?

10- The conclusion should be rephrased to show the recommended AgNPs concentration and to refer to future investigation.

The manuscript is well written

Author Response

Response to 2nd  Reviewer’s Comments

 (Manuscript ID: biomedicines-2509753)

  • The paper entitled: “In-vivo Bactericidal potential of Mangifera indica mediated silver nanoparticles against Aeromonas hydrophila in Cirrhinus mrigala” describes the efficacy of plant extract to fabricate AgNPs to biocontrol of bacterial infection caused by A. hydrophila into C. mrigala. The synthesized AgNPs were characterized by UV-vis spectroscopy, FTIR, XRD, SEM, and EDX. Moreover, the experimental design was achieved to investigate the efficacy of different concentrations of AgNPs on fish inoculated with pathogenic bacteria. The manuscript contains promising data and is well organized, but needs major revision before being accepted for publication in Biomedicines.
  • We are thankful to the learnt reviewer for expert assessment of our manuscript. We are obliged to the reviewer for mentioning ‘The manuscript contains promising data and is well organized’. All points and issues raised by the learnt reviewer have been considered and the manuscript has been modified accordingly. A detailed point-by-point response to all comments is provided below indicating the implemented changes in the revised version of the manuscript. A highlighted version by ‘Track Changes’ is also included with the resubmission
  • The hypothesis of the current study should be rephrased to be clearer.
  • We would like to express our sincere thanks to the reviewer for the valuable insights and the suggestions to improve the manuscript. We have clearly described hypothesis in the introduction section in revised version of the manuscript.

  • The novelty of the current study should be addressed.
  • Our study is novel as to the best of our knowledge there is no in vivo challenge study available on the anti-bacterial effect of mango leaves extract mediated AgNPs in Cirrhinus mrigala or any other carps. Although some in vitro assays against hydrophila have been performed but in vivo studies are lacking. Therefore, we believe that the findings of our study can open new era of the disease treatment in aquaculture thought plant mediated remedies and will help in minimizing the use of noxious antibiotics and other chemotherapeutic agents which by means of biomagnification in animal tissues can also be a health hazard to the humans.

According to reviewer suggestion the novelty of the data has been addressed in the main text in the conclusion section. 

  • The authors should refer to the efficacy of different biological entities in the green synthesis of Ag-NPs and their promising activities. The following references can be helpful: 

https://doi.org/10.3390/biom11030341; https://doi.org/10.3390/catal12050462 .

  • We are grateful to the learnt reviewer for guidance and hinting to the very relevant literature. The revise manuscript has been rectified according to the reviewer`s recommendations.

  • In Figure 1, please refer to the meaning of each letter in the Figure legend. This will be applied to each figure containing more panels.
  • We are thankful to the reviewer for highlighting this point, another reviewer also pointed out this issue. According to the reviewer’s recommendations, the alphabets (letters) have been removed from figure 1, 2 and 5. In all other figures, where alphabets are present, a brief explanation of each letter has been provided in the figure legends.

  • What about the optimization of pH value during NPs synthesis? Please clarify in subsection “2.1”
  • The Mi-AgNPs were prepared at different pH values and optimized value was found to be 7.6. The pH value of the solution was mentioned in the section 2.1, according to the reviewers’ recommendations.

  • In subsection “2.2. Characterization of biosynthesized Mi-AgNPs”, the characterization method should be discussed in detail. The following references can be helpful: https://doi.org/10.1016/j.heliyon.2020.e03943; https://doi.org/10.3390/ijms22105096  
  • We are appreciative the reviewer for the guidance and very relevant literature. In the ‘2.2. Characterization of biosynthesized Mi-AgNPs’ only very brief introduction including just name, make and models of characterization techniques are described. The details of characterization methods and related results of each characterization technique are presented in sections 3.1, 3.2 and 3.3. However, we have leant from the provided literature that how to present results in details and very impressive way and we have tried to rectified the revised manuscript in the light of these references. Once again, we are obliged.

  • In subsection “2.3. Experimental Organism and Conditions”, please clarify how to inoculate A. hydrophila into C. mrigala? Is the author standardized the inoculum?
  • For improving clarity of the method, detail description of hydrophilainduction into C. mrigala and the inocululam dose has been added in the in vivo challenge assay (section 2.4).

  • Please clarify why the authors achieved biochemicals and other analyses after 15 days of treatment. What happens if achieved after interval times?
  • From our prior hydrophila challenges studies [R1, R2], we had found that 15 days’ post infection (dpi) is a good time point for measuring the recovery effects of AgNPs on biochemical, hematological and histological parameters. In our previous 7dpi assays we found that changes had occurred but they were not significant as infection needed more time for progression and so the AgNPs. That is why we chose 15 dpi in this study. Moreover, we wanted to find the effects solely at the nonspecific/ innate immune system of the fish body being an important protective mechanism of the animal body. At the later stages (Approximately at 3 weeks dpi) the adaptive immune response (B and T cells) comes in play and calls another completely different repertoire of parameters to be involved.

However, we appreciate reviewer’s comment and would like to add particularly early time points along with 15 dpi for our future investigations.

  • How to detect the bacteria calculated in subsection 3.10 is A. hydrophila?
  • We detect the bacteria on the base of morphology analysis through microscope. Furthermore, we used LB agar and got yellowish colonies indication the presence of A. hydrophila. We use all the media, agar plates, and equipment which are sterilized to make sure there is no contamination during the plating method.

  • The conclusion should be rephrased to show the recommended AgNPs concentration and to refer to future investigation.
  • We are grateful to the reviewer for highlighting this point. The conclusion has been rephrased by adding the recommended best concentration from the present study and referred for future studies. The reviewer’s suggestion has clearly helped to make the conclusion more comprehensive.

  • The manuscript is well written
  • We want to express our sincere gratitude for the thoughtful and positive feedback on our work. Your appreciation of the manuscript means a great deal to us. Your insightful comments serve as a significant source of motivation for us.

We are again grateful to the reviewer for all the suggestions and recommendation which surely have extremely helped to improve our manuscript.

References:

  • Noshair, I.; Kanwal, Z.; Jabeen, G.; Arshad, M.; Yunus, F.-U.-N.; Hafeez, R.; Mairaj, R.; Haider, I.; Ahmad, N.; Alomar, S.Y. Assessment of Dietary Supplementation of Lactobacillus rhamnosus Probiotic on Growth Performance and Disease Resistance in Oreochromis niloticus. Microorganisms 2023, 11, 1423. https://doi.org/10.3390/microorganisms11061423
  • Pervaiz S., Kanwal Z., Manzoor F., Tayyeb A., Akram Q. Investigations on Blood Physiology, Tissues Histology and Gene Expression Profile of Fusarium oxysporum Challenged Fish. Sains Malays. 2022;51:2403–2414. doi: 10.17576/jsm-2022-5108-05.

Reviewer 3 Report

The manuscript contains valuable information. However, it needs some modifications as follows:

- Please provide the state-of-art. On the other hand, why such nanoparticles are useful and selected? What is the knowledge gap and novelty of the work?

- If possible, include DLS and zeta potential and XRD analysis.

- How did the authors calculate the exact yield of production of nanoparticles and the concentrations for the biological test?

- In the section of Discussion, the authors should also mention the factors that may have an influence on the biological activity of inorganic nanoparticles. These factors include size distribution, morphology, surface charge, surface chemistry, capping agents, etc. Follow and cite the articles below to support the above explanations.

https://doi.org/10.1016/j.talanta.2022.123374

https://doi.org/10.1016/B978-0-12-821013-0.00004-0

doi: 10.22037/ijpr.2020.113820.14504

Author Response

Response to 3rd Reviewer’s Comments

 (Manuscript ID: biomedicines-2509753)

  • The manuscript contains valuable information. However, it needs some modifications as follows:
  • We are thankful to the learnt reviewer for expert assessment of our manuscript. We are grateful to the reviewer for mentioning that ‘The manuscript contains valuable information’. Your appreciation of the manuscript means a great deal to us. Your insightful comments serve as a significant source of motivation for us. All points and issues raised by the learnt reviewer have been considered and the manuscript has been modified accordingly. A detailed point-by-point response to all comments is provided below indicating the implemented changes in the revised version of the manuscript. A highlighted version by ‘Track Changes’ is also included with the resubmission

  • Please provide the state-of-art. On the other hand, why such nanoparticles are useful and selected? What is the knowledge gap and novelty of the work?
  • The microbial activity of silver nanoparticles (AgNPs) is well-known. The green synthesis of lead to the facile, safe and ecofriendly approach for the preparation of AgNPs. The selection of Mangifera indica (mango) leaves were made on its importance as medicinal plant and antibacterial performance against fish pathogens [R1-R3]. The aim of this selection was to evaluate the potential combined effect of leaves extract and AgNPs against fish pathogen Aeromonas

Moreover, our study is novel as to the best of our knowledge there is no in vivo challenge study available on the anti-bacterial effect of mango leaves extract mediated AgNPs in Cirrhinus mrigala or any other carps. Although some in vitro assays against A. hydrophila have been performed but in vivo studies are lacking. Therefore, we believe that the findings of our study can open new era of the disease treatment in aquaculture thought plant mediated remedies and will help in minimizing the use of noxious antibiotics and other chemotherapeutic agents which by means of biomagnification in animal tissues can also be a health hazard to the humans. 

  • If possible, include DLS and zeta potential and XRD analysis.
  • We are grateful to the reviewer for pointing out these important analysis, DLS (dynamic light scattering) and zeta potential facilities are not available in our labs. We will consider them in our future research. However, XRD analysis is already included in the manuscript as discussed in Figure 3 (b).

  • How did the authors calculate the exact yield of production of nanoparticles and the concentrations for the biological test?
  • As the results of extract mediated synthesis, colloidal samples of Mi-AgNPs were obtained, from these sample pallets of the Mi-AgNPs were obtained by literature reported methods [R4]. From these he desired concentration were made accordingly for biological tests.

  • In the section of Discussion, the authors should also mention the factors that may have an influence on the biological activity of inorganic nanoparticles. These factors include size distribution, morphology, surface charge, surface chemistry, capping agents, etc. Follow and cite the articles below to support the above explanations.

https://doi.org/10.1016/j.talanta.2022.123374

https://doi.org/10.1016/B978-0-12-821013-0.00004-0

doi: 10.22037/ijpr.2020.113820.14504

  • We are thankful the reviewer for the guidance and literature, the discussion has been rectified according to the reviewer recommendations.

At the end we want to express our sincere gratitude to the reviewer for the thoughtful comments and feedback. All the suggestions and recommendation given by the reviewer have surely helped a lot to improve our manuscript.

References:

  • Sahu, S.; Das, B.K.; Pradhan, J.; Mohapatra, B.C.; Mishra, B.K.; Sarangi, N. Effect of Magnifera indica kernel as a feed additive on immunity and resistance to Aeromonas hydrophila in Labeo rohita fingerlings, Fish Shellfish Immunol. 2007, 23(1),109-118, https://doi.org/10.1016/j.fsi.2006.09.009 .
  • Alaiya, M.A., Odeniyi, M.A. Utilisation of Mangifera indica plant extracts and parts in antimicrobial formulations and as a pharmaceutical excipient: a review. J. Pharm. Sci. 2023, 9, 29. https://doi.org/10.1186/s43094-023-00479-z
  • Maharaj, A.; Naidoo, Y.; Dewir, Y.H.; Rihan, H. Phytochemical Screening, and Antibacterial and Antioxidant Activities of Mangifera indica Leaves. Horticulturae 2022, 8, 909. https://doi.org/10.3390/horticulturae8100909
  • Badar, W.; Ullah Khan, M.A. Analytical study of biosynthesised silver nanoparticles against multi-drug resistant biofilm-forming pathogens. IET Nanobiotechnol. 2020, 14(4), 331-340. https://doi.org/10.1049/iet-nbt.2019.0287.

Round 2

Reviewer 1 Report

The authors have modified the manuscript according to the comments given.

Author Response

We are thankful to reviewer for acknowledgment of our response to the comments and satisfaction on the revised manuscript.

Reviewer 2 Report

The manuscript is suitable for publication in the current form

The language is satisfying 

Author Response

We are obliged to the reviewer for recommending our revised manuscript for publication.